# Dual role of FOXG1 in regulating gliogenesis in the developing neocortex via the FGF signalling pathway

**Mahima Bose[1], Ishita Talwar[1†], Varun Suresh[1†], Urvi Mishra[1†], Shiona Biswas[1], Anuradha Yadav[1], Shital T Suryavanshi[1], Simon Hippenmeyer[2], Shubha Tole[1]\***

[1]Department of Biological Sciences, Tata Institute of Fundamental Research, Mumbai, India; [2]Institute of Science and Technology Austria, Klosterneuburg, Austria

## eLife Assessment

This **important** study provides **convincing** evidence that developing neurons in the neocortex regulate glial cell development. The data demonstrates that the transcription factor FOXG1 negatively regulates gliogenesis by controlling the expression of a member of the FGF ligand family and by suppressing the receptor for this ligand in developing neurons. This study leads to a new understanding of the cascade of events regulating the timing of glial development in the neocortex.

**\*For correspondence:**
shubhatole@gmail.com

†These authors contributed equally to this work

**Competing interest:** The authors declare that no competing interests exist.

**Abstract** In the developing vertebrate central nervous system, neurons and glia typically arise sequentially from common progenitors. Here, we report that the transcription factor Forkhead Box G1 (*Foxg1*) regulates gliogenesis in the mouse neocortex via distinct cell-autonomous roles in progenitors and postmitotic neurons that regulate different aspects of the gliogenic FGF signalling pathway. We demonstrate that loss of *Foxg1* in cortical progenitors at neurogenic stages causes premature astrogliogenesis. We identify a novel FOXG1 target, the pro-gliogenic FGF pathway component *Fgfr3*, which is suppressed by FOXG1 cell-autonomously to maintain neurogenesis. Furthermore, FOXG1 can also suppress premature astrogliogenesis triggered by the augmentation of FGF signalling. We identify a second novel function of FOXG1 in regulating the expression of gliogenic cues in newborn neocortical upper-layer neurons. Loss of FOXG1 in postmitotic neurons non-autonomously enhances gliogenesis in the progenitors via FGF signalling. These results fit well with the model that newborn neurons secrete cues that trigger progenitors to produce the next wave of cell types, astrocytes. If FGF signalling is attenuated in *Foxg1* null progenitors, they progress to oligodendrocyte production. Therefore, loss of FOXG1 transitions the progenitor to a gliogenic state, producing either astrocytes or oligodendrocytes depending on FGF signalling levels. Our results uncover how FOXG1 integrates extrinsic signalling via the FGF pathway to regulate the sequential generation of neurons, astrocytes, and oligodendrocytes in the cerebral cortex.

## Introduction

A fundamental feature of the developing vertebrate central nervous system is that common progenitors produce neurons and glia sequentially. Furthermore, two broad categories of glia, astrocytes, and oligodendrocytes, each sub-serving distinct functions, are also produced sequentially. This temporal sequence of neurogenesis, followed by astrogliogenesis and then oligogenesis, is conserved in the developing mammalian neocortex (*Qian et al., 2000*; *Hirabayashi et al., 2009*; *Reynolds and Weiss, 1992*). However, the molecular mechanisms instructing a cessation of neurogenesis and initiating

gliogenesis within progenitors are poorly understood, particularly in terms of how cell-intrinsic factors interact with cell-extrinsic signalling cues.

Early in vitro (*Qian et al., 2000*; *Temple, 1989*) and in vivo (*Shen et al., 2021*) studies demonstrated that cortical progenitors first produce neurons and later glia. Quantitative Mosaic Analysis with Double Markers (MADM) lineage analyses in vivo further extended these findings and showed that neurons, astrocytes, and oligodendrocytes arise sequentially within a clone of cells from a single progenitor (*Shen et al., 2021*; *Gao et al., 2014*). Molecular insights into this phenomenon identified several pro-gliogenic mechanisms. These include cell-intrinsic transcription factors *Nr2f1/2*, *Zbtb20*, *Sox9*, and *Nfia* (*Naka et al., 2008*; *Namihira et al., 2009*; *Kang et al., 2012*; *Nagao et al., 2016*), the Notch pathway (*Namihira et al., 2009*; *Tanigaki et al., 2001*; *Cao et al., 2008*; *Ge et al., 2002*; *Cao et al., 2010*); and cell-extrinsic cues such as interleukins IL6 and cardiotrophin1 (*Ct1*), leukemia inhibiting factor (*Lif*), ciliary neurotrophic factor (*Cntf*), neuropoietin (*Np*), cardiotrophin-like cytokine (*Clc*), epidermal growth factors (*Egf*), and fibroblast growth factors (*Fgf*; *Rowitch and Kriegstein, 2010*; *Mallamaci, 2013*; *Sloan and Barres, 2014*; *Takouda et al., 2017*; *Dinh Duong et al., 2019*; *Zhang et al., 2020*), which act via the Jak/Stat or Mek/Mapk signalling pathways (*Cao et al., 2008*; *Nakashima et al., 1999*; *He et al., 2005*). In these studies, the primary readout of gliogenesis was astrogliogenesis, using either morphology or molecular markers to identify glial fate. Although a few factors, such as *Sox10,* have been reported to promote oligodendrocyte precursor cell (OPC) fate (*Stolt et al., 2002*; *Bhattarai et al., 2022*), the progenitor-level mechanisms that govern astrocyte-to-oligodendrocyte transition remain poorly understood. Furthermore, how mechanisms that maintain ongoing neurogenesis crosstalk with those that promote gliogenesis remains to be understood. Another unexplored area is how cell-intrinsic and extrinsic factors interact with or regulate each other at the level of the individual progenitor.

Forkhead family transcription factor FOXG1 plays a fundamental role in the early development of the cerebral cortex. It is a well-established regulator of the sequential production of neuronal subtypes in the mammalian neocortex (*Hanashima et al., 2002*; *Toma et al., 2014*; *Hanashima et al., 2004*; *Kumamoto and Hanashima, 2017*; *Hou et al., 2020*; *Liu et al., 2022*; *Tao and Lai, 1992*). *Foxg1* haploinsufficiency in humans affects a range of neurodevelopmental processes, resulting in an autism spectrum disorder called Foxg1 syndrome, which includes corpus callosum agenesis, microcephaly, and cognitive impairment (*Shoichet et al., 2005*). A dysregulation of *Foxg1* has also been implicated in glioblastoma pathogenesis among these individuals (*Hou et al., 2020*; *Hettige and Ernst, 2019*).

FOXG1 is described to play a 'neurogenic role' in overexpression experiments in vitro or cultured progenitors transplanted in vivo, which display decreased gliogenesis and downregulation of progliogenic pathways and markers (*Brancaccio et al., 2010*; *Falcone et al., 2019*; *Frisari et al., 2022*). However, in these studies, *Foxg1* knockdown did not cause significant gliogenesis (*Brancaccio et al., 2010*; *Falcone et al., 2019*). Furthermore, the role of FOXG1 in postmitotic neurons in regulating gliogenesis has not been examined, although these cells express *Foxg1* and are ideally positioned to signal progenitors to generate the next cell type. These are critical unanswered questions that we explored directly in vivo. We show that upon loss of *Foxg1*, progenitors autonomously produce glia at the expense of neurons. Using integrative analysis of the transcriptome, epigenome, and FOXG1 occupancy, we identified FGFR3, part of the established progliogenic FGF signalling pathway, as a FOXG1 target.

We show that the loss of *Foxg1* results in upregulating *Fgfr3* expression and FGF signalling in cortical progenitors. Whereas overexpression of FGF ligands induces premature gliogenesis (*Dinh Duong et al., 2019*), simultaneous overexpression of *Foxg1* restores neurogenesis, consistent with its established neurogenic role. Furthermore, attenuation of FGF signalling in a *Foxg1* loss-of-function (LOF) background leads to a premature production of OPCs, indicating that without FOXG1 the progenitor may progress through gliogenesis but cannot return to neurogenesis.

Independently, FOXG1 regulates cues produced by newborn postmitotic neurons that non-autonomously modulate the output of progenitors. In particular, when postmitotic neurons lack *Foxg1*, naive progenitors upregulate FGF signalling and display enhanced gliogenesis. We demonstrate that progenitors experience an enhanced level of FGF signalling when *Foxg1* is lost selectively in postmitotic neurons due to dysregulated *Fgf* ligand expression. Therefore, FOXG1 regulates both the availability of the ligand and the level of the receptor for Fgf signalling.

Our results shed light on a novel mechanism regulating the sequential generation of neurons, astrocytes, and oligodendrocytes in the cerebral cortex.

## Results

We used a *Foxg1* conditional knockout mouse line with an intrinsic reporter (*Foxg1*<sup>lox/lox</sup>; *Rosa26*<sup>FRT-GFP</sup>; *Miyoshi and Fishell, 2012*) to assess the cell-autonomous phenotypes of *Foxg1* disruption. In this line, Cre-mediated recombination of the floxed *Foxg1* allele results in the expression of flippase, which then recombines a STOP-FRT-EGFP reporter inserted in the *Rosa26* locus. Since this reporter is integrated into the genome, introducing Cre recombinase into a progenitor results in GFP expression in *Foxg1*-recombined cells arising from it without dilution, serving as a label for its lineage.

### Disrupting *Foxg1* during neurogenesis causes premature gliogenesis

In wild-type mice, FOXG1 expression decreases in cortical progenitors from embryonic day (E) 15.5 to E18.5 (*Figure 1—figure supplement 1*; *Falcone et al., 2019*). We tested the consequences of a premature drop in FOXG1 at E15.5 when superficial layer neurons are generated in the neocortex. We introduced a plasmid encoding Cre recombinase at E15.5 using in utero electroporation and examined the brains on postnatal day 14 (P14). This stage is suitable for a comprehensive assessment of the lineage arising from control and mutant progenitors.

In control brains, as expected, GFP+ cells with pyramidal neuron morphologies occupied layer II/III of the neocortex, whereas cells with glial morphologies were scattered throughout the tissue. 67% of the GFP+ cells also expressed neuronal marker NEUN and did not express mature astrocytic marker ALDH1L1. 16% of the GFP+ cells were also ALDH1L1+ due to the intrinsic *Rosa26*<sup>FRT-GFP</sup> reporter that causes EGFP expression in the entire lineage of progenitors electroporated at E15.5 (*Figure 1A–D*, *Figure 1—figure supplement 2*). In contrast, Cre electroporation in *Foxg1*<sup>lox/lox</sup> embryos results in GFP+ cells with astrocytic morphologies distributed all over the neocortex. None were NEUN+, but 56% were ALDH1L1+, confirming their astrocyte identity. We found similar results, that is, premature gliogenesis, upon Cre electroporation at E14.5 (*Figure 1E–H*). In contrast, loss of *Foxg1* at E13.5 did not impede neurogenesis. The mutant cells were NEUN+ and ALDH1L1−, indicating no premature gliogenesis. Consistent with the literature, there is enhanced expression of REELIN (*Figure 1—figure supplement 3*; *Toma et al., 2014*; *Hanashima et al., 2004*; *Shen et al., 2006*; *Kumamoto and Hanashima, 2014*; *Kumamoto et al., 2013*).

This indicates that progenitors undergo a transition after E13.5, wherein the loss of FOXG1 from E14.5 prompts a shift from neurogenesis to gliogenesis, even if they have not yet produced the final cohorts of superficial layer neurons.

### Enhanced gliogenesis upon loss of *Foxg1* is not due to the over-proliferation of mutant astrocytes

Increased gliogenesis may arise from the over-proliferation of astrocytes and an accompanying death of neurons that have lost *Foxg1*. Alternatively, there could be a premature cell fate switch in the progenitors that should have produced neurons but instead produced astrocytes (*Figure 2A*). To distinguish between these scenarios, we examined the presence of KI67 in cells at 1 day (E16.5) or 3 days (E18.5) after Cre electroporation at E15.5. Similar numbers of electroporated (EGFP+) cells displayed KI67 in control and *Foxg1*<sup>lox/lox</sup> brains, indicating that loss of *Foxg1* did not induce over-proliferation (*Figure 2B and B'*, *Figure 2—figure supplement 1*). Furthermore, no difference was observed in the expression of a cell death marker Cleaved Caspase 3 (*Figure 2C and C'*). However, consistent with a switch to gliogenesis, the *Foxg1* LOF cells were positive for gliogenic factor NFIA but not the neurogenic intermediate progenitor marker TBR2 (*Figure 2D, D', E and E'*).

We also employed a MADM-based strategy to distinguish between these possibilities (*Figure 2F*; *Zong et al., 2005*; *Contreras et al., 2021*). We used the MADM-12 (M12) line (*Contreras et al., 2021*; *Hippenmeyer et al., 2013*) in which the MADM GT and TG cassettes are on chromosome 12 and crossed them onto a *Foxg1*<sup>lox/+</sup> background. In the resulting *Foxg1*-MADM (*M12*<sup>GT/TG, Foxg1</sup>) brains, a fraction of Cre-electroporated progenitors undergo recombination, resulting in progeny that is either *Foxg1*<sup>−/−</sup>, GFP+ (green), or *Foxg1*<sup>+/+</sup> tdTomato+ (red) cells. Since these red and green cells arise from single progenitors, the MADM system enables an accurate analysis of proliferation versus

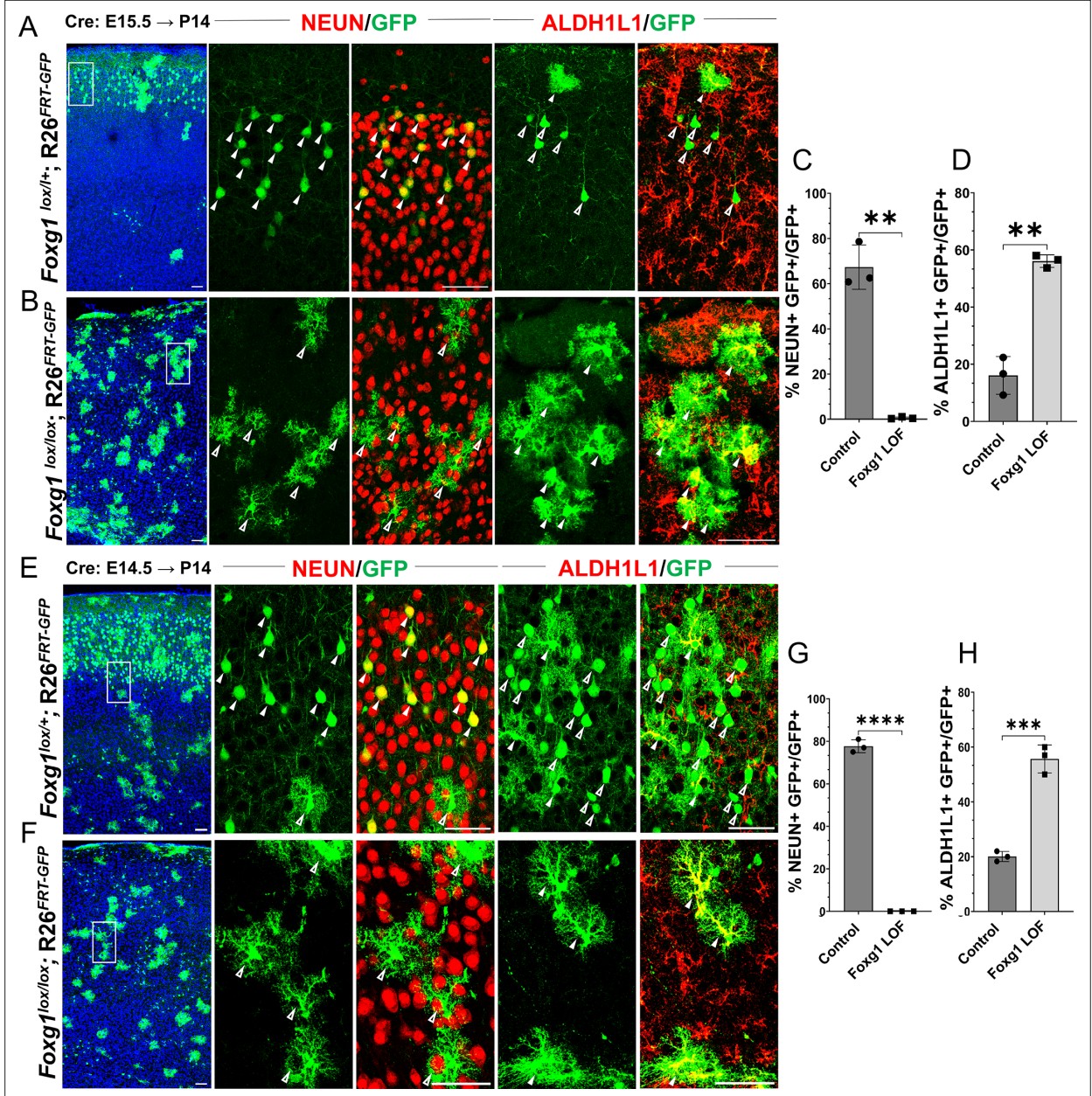

**Figure 1.** Loss of *Foxg1* causes premature gliogenesis. (**A–D**) Cre electroporation at E15.5 in control (**A**, *Foxg1*[lox/+]; *Rosa26*[FRT-GFP]) and *Foxg1* loss-of-function (LOF) (**B**, *Foxg1*[lox/lox]; *Rosa26*[FRT-GFP]) embryos, followed by analysis at P14. (**C, D**) 67.3% of GFP+ cells colocalised with NeuN in control brains and 1% in *Foxg1* LOF brains. 16% of GFP+ cells colocalised with ALDH1L1 in control brains and 56% in *Foxg1* LOF brains. n = 2151 (Control), 2761 (*Foxg1* LOF) cells from N = 3 brains (biologically independent replicates). (**E–H**) Cre electroporation at E14.5 in control (**E**, *Foxg1*[lox/+]; *Rosa26*[FRT-GFP]) and *Foxg1* LOF (**F**, *Foxg1*[lox/lox]; *Rosa26*[FRT-GFP]) embryos, followed by analysis at P14. (**G, H**) 77.7% of GFP+ cells colocalised with NeuN in control brains and 0% in *Foxg1* LOF brains. 20.1% of GFP+ cells colocalised with ALDH1L1 in control brains and 55.7% in *Foxg1* LOF brains. n = 3,160 (Control), 2978 (*Foxg1* LOF) cells from N = 3 brains (biologically independent replicates). In each row (**A, B, E, F**), the boxes in the leftmost low magnification panels indicate approximate regions shown in either the NEUN or ALDH1L1 high-magnification panels. Filled arrowheads depict colocalisation, and open arrowheads depict non-colocalisation of marker and electroporated cells. Statistical test: two-tailed unpaired *t*-test. *p<0.05, **p<0.01, ***p<0.001, ****p<0.0001. All scale bars are 50 µm.

The online version of this article includes the following figure supplement(s) for figure 1:

**Figure supplement 1.** Complementary temporal regulation of *Foxg1* and *Fgfr3* in progenitors during the gliogenic period.

**Figure supplement 2.** *Foxg1* haploinsufficiency at E15.5 does not lead to premature gliogenesis.

**Figure supplement 3.** Loss of *Foxg1* at E13.5 does not lead to premature gliogenesis but results in increased REELIN.

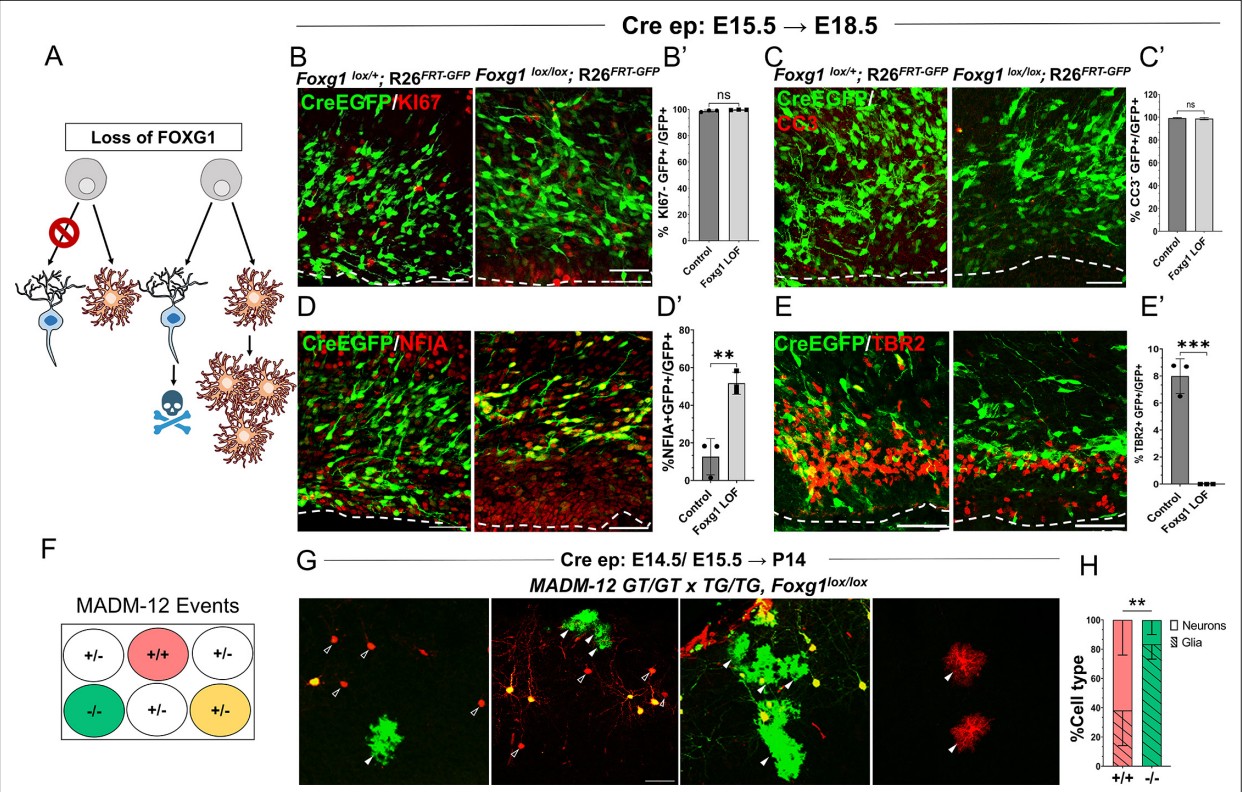

**Figure 2.** *Foxg1* loss-of-function (LOF) leads to premature glial fate acquisition in progenitors but no proliferation defects. (**A**) Schematic depicting the two alternative mechanisms that could result in enhanced gliogenesis upon loss of *Foxg1*: a change in cell type produced by the progenitor, i.e., 'fate switch' or increase in proliferation of astrocytes accompanied by neuronal cell death. (**B, B', C, C'**) Images of the ventricular and sub-ventricular zone (the dashed line indicates the ventricle boundary). Cre electroporation at E15.5 in control (*Foxg1*^lox/+^; *Rosa26*^FRT-GFP^) and *Foxg1* LOF (*Foxg1*^lox/lox^; *Rosa26*^FRT-GFP^) embryos, followed by analysis at E18.5. Proliferation marker KI67 (**B, B'**) colocalises with similar numbers of GFP+ cells in control and *Foxg1* LOF brains. Cell death marker Cleaved Caspase 3 (**C, C'**) does not reveal differences in colocalisation with GFP+ control and *Foxg1* LOF cells. In contrast, glial progenitor markers NFIA (**D, D'**) display increased colocalisation with GFP+ cells in *Foxg1* LOF (51.7%) compared with controls (12.5%). Neurogenic intermediate progenitor marker TBR2 (**E, E'**) displays decreased colocalisation with GFP+ cells in *Foxg1* LOF brains (0%) compared with controls (8%). n = 3590 (control), 2100 (mutant) cells from N = 3 brains (biologically independent replicates). (**F**) Schematic depicting the genotype and corresponding fluorescent labels resulting from the Mosaic Analysis with Double Markers (MADM) recombination events. (**G-H**) Cre electroporation at E14.5/E15.5 in *Foxg1*-MADM brains (*M12*^GT/TG, Foxg1^) analysed at P14. Green (*Foxg1*^−/−^) and red (*Foxg1*^+/+^) cells were scored based on neuronal (open arrowheads, **G**) or glial (arrowheads, **E**) morphology. (**H**) Represents the number of neurons or glia as a percentage of the total population of neurons +glia of each genotype: control (red; +/+) or *Foxg1* mutant (green –/–) neurons. n = 354 cells from N = 5 brains (biologically independent replicates). Statistical test: two-tailed unpaired *t*-test. *p<0.05, **p<0.01, ***p<0.001, ****p<0.0001. All scale bars: 50 µm.

The online version of this article includes the following figure supplement(s) for figure 2:

**Figure supplement 1.** Loss of *Foxg1* from E15.5 does not lead to enhanced proliferation at E16.5.

cell fate transformation upon disruption of *Foxg1*. A fraction of progenitors in the MADM paradigm undergoes a different recombination pattern, resulting in a yellow progeny due to the expression of both GFP and tdTomato (*Zong et al., 2005*; *Contreras et al., 2021*), which were not scored. We performed in utero electroporation of Cre in *Foxg1*-MADM embryos in E14.5/E15.5 embryos. We analysed the brains at P14 (*Figure 2G*). Green and red cells were scored for neuronal or glial identity based on morphology (*Shen et al., 2021*; *Zhang et al., 2020*). Of the total control (red, +/+) cells, 61.9% were neurons, and 38.1% were glia. In contrast, of the total number of *Foxg1* mutant (green,–/–) cells, 16.7% were neurons, and 83.3% were glia (*Figure 2H*).

In summary, the results demonstrate that loss of *Foxg1* results in a premature cell fate switch in neurogenic progenitors, making them gliogenic.

## Loss of *Foxg1* causes upregulation of *Fgfr3* in progenitors and enhanced FGF signalling

We investigated how FOXG1 regulates the transcriptional landscape in cortical progenitors to maintain a temporal control of neurogenesis and gliogenesis. We used the inducible hGFAP-CreERT2 (hGCE) driver to achieve widespread recombination in apical progenitors at embryonic ages upon induction with tamoxifen (*Zhuo et al., 2001*; *Ganat et al., 2006*). First, we ascertained that hGCE-driven loss of Foxg1 at E15.5 recapitulates the premature gliogenesis phenotype we described in *Figure 3—figure supplement 1A*. Also, 2 days after tamoxifen induction (E15.5 → E17.5), *Foxg1* LOF cells were positive for gliogenic factor NFIA, similar to that seen upon Cre electroporation (*Figure 2C*, *Figure 3— figure supplement 1B and C*). To identify pathways that regulate the premature gliogenesis induced by the loss of *Foxg1*, we collected FACS-purified control and mutant cells at E17.5, 48 hours post-tamoxifen administration at E15.5, for transcriptomic analysis (*Figure 3—figure supplement 1D and*

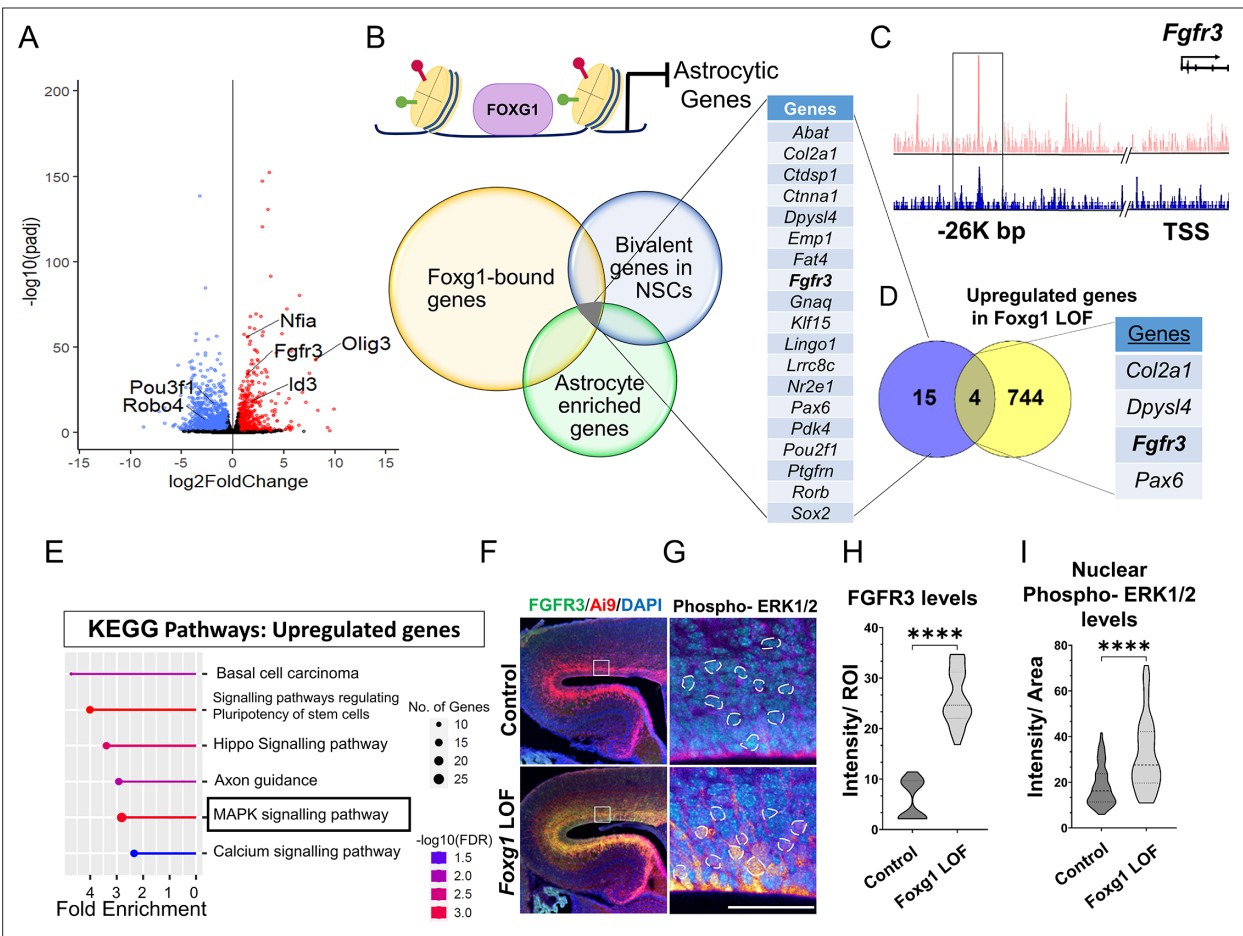

**Figure 3.** FOXG1 binds and regulates the expression of *Fgfr3*. (**A**) RNA-seq analysis of FACS-purified Control and *Foxg1* loss-of-function (LOF) progenitors harvested two days after labelling at E15.5. Gliogenic factors such as *Nfia, Id3,* and *Olig3* are upregulated, and neuronal markers such as *Pou3f1* and *Robo4* are downregulated. (**B–D**) Multimodal analysis comparing FOXG1 occupancy (ChIP-seq) and bivalent epigenetic marks (H3K4Me3 and H3K27Me3) and astrocyte-enriched genes from *Telley et al., 2019* reveals a list of 19 genes common to each dataset (**B**). Four of these are upregulated upon loss of *Foxg1*, including the known gliogenic gene *Fgfr3* (**D**). FOXG1 occupies a −26 kb enhancer region of *Fgfr3* (**C**). (**E**) KEGG analysis of the upregulated genes from (**A**) identifies the MAPK signalling pathway downstream of FGF signalling. (**F**) Loss of *Foxg1* from E15.5 progenitors (hGFAP-CreERT2, tamoxifen at E15.5) causes upregulation of FGFR3 receptor by E17.5 as seen in cells near the VZ of the somatosensory cortex. Boxes (**F**) indicate the regions in high magnification shown in the adjacent panels (**G**). Dashed circles outline the regions of interest (ROIs) identified in the DAPI channel used for intensity quantification in (**H**). (**G**; n = 50 [Control and *Foxg1* LOF] ROIs from N = 3 brains) and phosphorylated-ERK1/2 (**H**; n = 67 [Control] and 89 [*Foxg1* LOF]) cells from N = 3 brains (biologically independent replicates). Statistical test: Mann–Whitney test *p<0.05, **p<0.01, ***p<0.001, ****p<0.0001. All scale bars: 50 µm.

The online version of this article includes the following figure supplement(s) for figure 3:

**Figure supplement 1.** hGFAP-CreERT2; Ai9 (hGCE) line-based recapitulation of the premature gliogenesis phenotype and subsequent analysis.

*D'*; *Supplementary file 1*). As expected, the loss of *Foxg1* caused the upregulation of known glio-genic factors such as *Nfia* and *Olig3* and the downregulation of neurogenic genes such as *Neurod2* and neuronal marker *Pou3f1* (*Figure 3A*).

We further filtered the list of genes upregulated upon loss of *Foxg1* using a strategy designed to narrow down candidates likely to drive gliogenesis. First, we examined a set of genes known to display FOXG1 occupancy at E15.5 (*Cargnin et al., 2018*). Second, we reasoned that cell fate transitions are driven by genes maintained in states poised for transcriptional activation or repression by displaying both epigenetic marks (*Liu et al., 2017*). Therefore, we analysed a dataset from E15.5 radial glial cells to identify genes that carry bivalent H3K4Me3 and H3K27Me3 marks in gene regulatory regions. Third, we obtained a list of genes enriched in cortical astrocytes (*Telley et al., 2019*).

The overlap of these three datasets yielded a shortlist of 19 genes representing potential FOXG1 targets that may be suppressed during neurogenesis (*Figure 3B and C*). Finally, we compared this multimodal analysis with the list of genes we discovered to be upregulated upon loss of *Foxg1* and identified four targets: *Col2a1, Dpysl4, Fgfr3*, and *Pax6* (*Figure 3D*). The presence of *Fgfr3* in this set presented the exciting possibility that FOXG1 may function to suppress the potent progliogenic FGF signalling pathway (*Dinh Duong et al., 2019*) in neurogenic cortical progenitors.

In summary, we identified *Fgfr3* as a novel FOXG1 target. Upregulation of *Fgfr3* may mediate the gliogenic effects of loss of *Foxg1*. Consistent with this hypothesis, the MAPK/ERK pathway, which mediates FGF signalling, was identified as a prominent upregulated pathway in a KEGG pathway analysis of the RNA-seq data (*Figure 3E*). Activation of this pathway results in the phosphorylation of ERK1/2 and its translocation to the nucleus to regulate the downstream targets of FGFR activation (*Lavoie et al., 2020*). To assess whether the loss of *Foxg1* indeed has functional consequences on Fgf signalling, we examined FGFR3 and nuclear phospho-ERK1/2 levels in control and *Foxg1* mutant cortical progenitors. Loss of *Foxg1* at E15.5 led to an increase in FGFR3 labelling and nuclear locali-sation of phospho-ERK1/2 by E17.5 (*Figure 3F–H*). We quantified nuclear FOXG1 levels in wild-type apical progenitors. We found a significant decrease from E15.5 to E18.5, consistent with previous findings (*Falcone et al., 2019*), suggesting an endogenous mechanism for *Fgfr3* upregulation with the initiation of gliogenesis (*Figure 1—figure supplement 1*).

These results demonstrate that FOXG1 suppresses *Fgfr3* gene expression and the MAPK/ERK pathway, and loss of *Foxg1* results in an enhancement of this pro-gliogenic pathway. The decline in nuclear FOXG1 by E18.5 suggests an endogenous mechanism for the transition of neurogenesis to gliogenesis in apical progenitors.

## *Foxg1* overexpression cell-autonomously suppresses FGF-induced astrogliogenesis

Exogenous FGF8 is known to have potent progliogenic effects in the neocortex *Dinh Duong et al., 2019*, which our results recapitulated (*Figure 4A and B*). As expected, electroporation at E15.5 of a control construct encoding EGFP did not impede neurogenesis. In wild-type mice, 100% of the electroporated GFP+ cells were NEUN+ and occupied the superficial layers of the neocortex by P7 (*Figure 4A*). Similar to the findings of *Dinh Duong et al., 2019*, co-electroporation of constructs encoding *Fgf8+Egfp* induced premature gliogenesis, resulting in 85% of the GFP+ cells displaying astrocytic morphologies and glial marker SOX9 (*Figure 4B*). *Foxg1* overexpression alone did not affect neurogenesis but gave rise to neurons, some of which occupied the superficial layer and some displayed migration deficits, as previously shown (*Miyoshi and Fishell, 2012*; *Figure 4C*). Note that this experiment did not use the intrinsic *Rosa26^{FRT-GFP}* reporter. Electroporation-based introduction of GFP is expected to dilute with continued progenitor proliferation. Therefore, the control brains displayed GFP in neurons that were born at E15.5, and not in glia that arose subsequently from the same progenitors.

In summary, FOXG1 is sufficient to cell-autonomously suppress the pro-gliogenic effects of FGF8, establishing it as a regulator of FGF signalling within cortical progenitors.

## Postmitotic neuron-specific loss of *Foxg1* leads to premature gliogenesis in cortical progenitors

Postmitotic neurons provide 'feedback instruction' to progenitors via cues that modulate the cell types that arise subsequently (*Seuntjens et al., 2009*; *Barnabé-Heider et al., 2005*). We examined

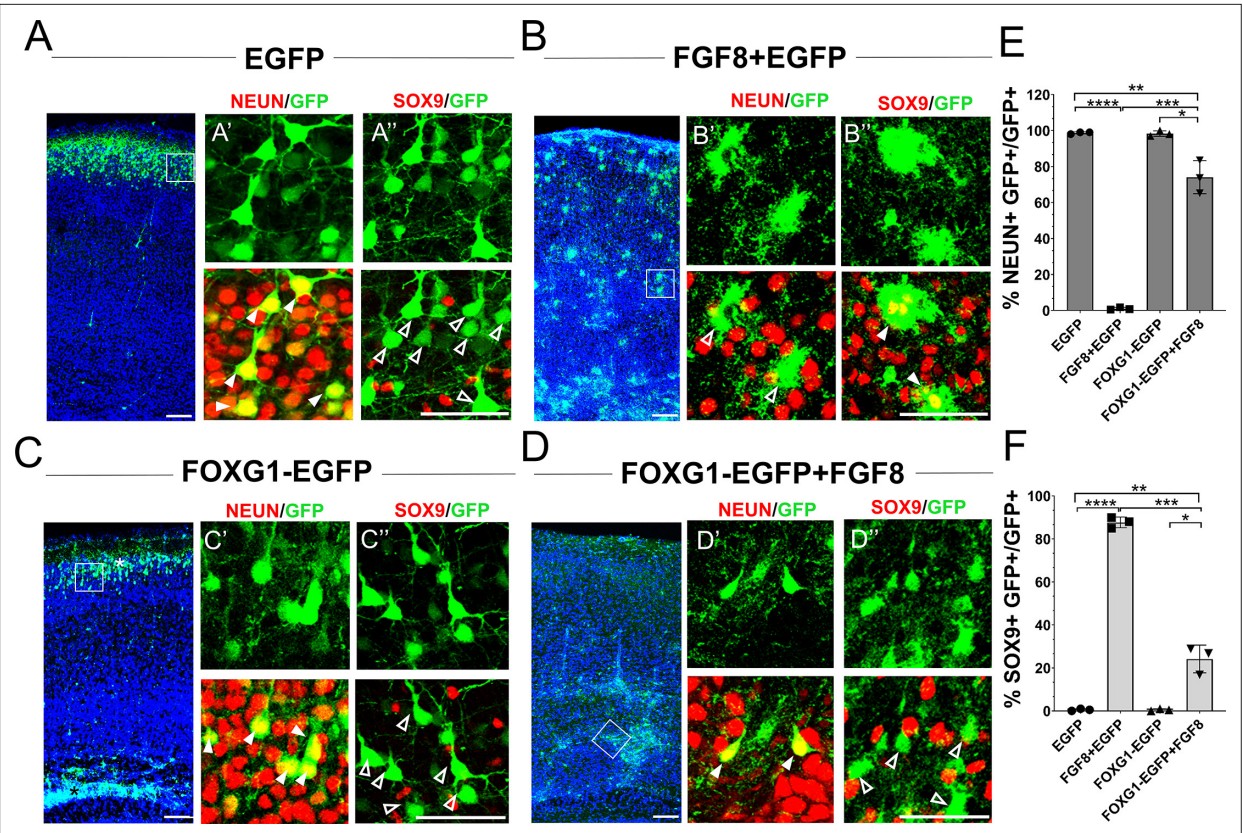

**Figure 4.** *Foxg1* suppresses FGF-induced gliogenesis. (**A–D**) In-utero electroporations were performed in wild-type embryos at E15.5, and the brains were analysed at P7. (**A**) GFP electroporation labels LII/III cells that are NEUN+ (arrowheads) and SOX9– (open arrowheads). (**B**) Overexpression of *Fgf8* leads to premature gliogenesis, and the GFP+ cells are NEUN– (open arrowheads) and SOX9+ (arrowheads). (**C**) Overexpression of *Foxg1* produced NEUN+ (arrowheads) and SOX9– (open arrowheads) neurons, some of which displayed delayed migration (black asterisk), and others migrated to the cortical plate (white asterisk), as shown in *Miyoshi and Fishell, 2012*. (**D**) Overexpression of *Foxg1* together with FGF8 partially rescued neuronal fate such that GFP+ cells also displayed NEUN (arrowheads) but not SOX9 (open arrowheads). In (**A–D**), the boxes in the leftmost low-magnification panels indicate approximate regions shown in the adjacent high-magnification panels. (**E**) Quantifications of GFP+ cells that are also NEUN+ in each condition: 98.6% (GFP); 1.8% (*Fgf8*); 98.3% (*Foxg1*); 74.1% (*Foxg1+Fgf8*). (**F**) Quantifications of GFP+ cells that are also SOX9+ in each condition: 0% (*Egfp*); 87.7% (*Fgf8+Egfp*); 0% (*Foxg1-Egfp*); 24.2% (*Foxg1-Egfp+Fgf8*). n = 2,123 (*Egfp*), 1643 (*Fgf8+Egfp*), 1357 (*Foxg1-Egfp*), 1924 (*Foxg1-Egfp+Fgf8*) cells each from N = 3 brains (biologically independent replicates). Statistical test: two-way ANOVA with Tukey's correction. *p<0.05, **p<0.01, ***p<0.001, ****p<0.0001. All scale bars: 50 µm.

a publicly available transcriptomic dataset from CUX2+ upper-layer neurons isolated at stages from E18.5 to P48 (*Yuan et al., 2022*) and discovered that Fgf family members *Fgf9, Fgf10,* and *Fgf18* mRNA levels peaked in the first postnatal week and dropped thereafter, consistent with the temporal profile of astrogliogenesis in the cortex (*Figure 5—figure supplement 1A and B*; *Thompson et al., 2014*). Since *Foxg1* is expressed in postmitotic neurons (*Hou et al., 2020*; *Dastidar et al., 2011*), we tested whether it may play a role in regulating gliogenic factors secreted by these cells. We used post-mitotic neuron-specific *NexCre* to disrupt *Foxg1* and examined these brains at birth (*Figure 5—figure supplement 1C and D*). We found a significant increase in the levels of gliogenic factor SOX9 in the ventricular zone progenitors and an increased number of SOX9+ cells in the mantle compared to that in control brains, indicating a non-autonomous effect of neuron-specific loss of *Foxg1* on progenitors (*Figure 5A–D*). Consistent with enhanced gliogenesis, there was an apparent increase in astrocyte marker ALDH1L1+ cells in the entire cortical plate (*Figure 5—figure supplement 1E*).

To identify potential gliogenic molecules regulated by FOXG1 in postmitotic neurons, we performed transcriptomic analysis on cortical plate tissue isolated from control and *NexCre*-driven *Foxg1* LOF brains at birth and examined genes encoding secreted factors (*Figure 5E*, *Supplementary file 2*). In the Fgf family, *Fgf9* transcripts decreased, *Fgf10* displayed no change, whereas *Fgf18* displayed a 2.5-fold increase (*Figure 5E*). This ligand has been well characterised to be preferentially expressed

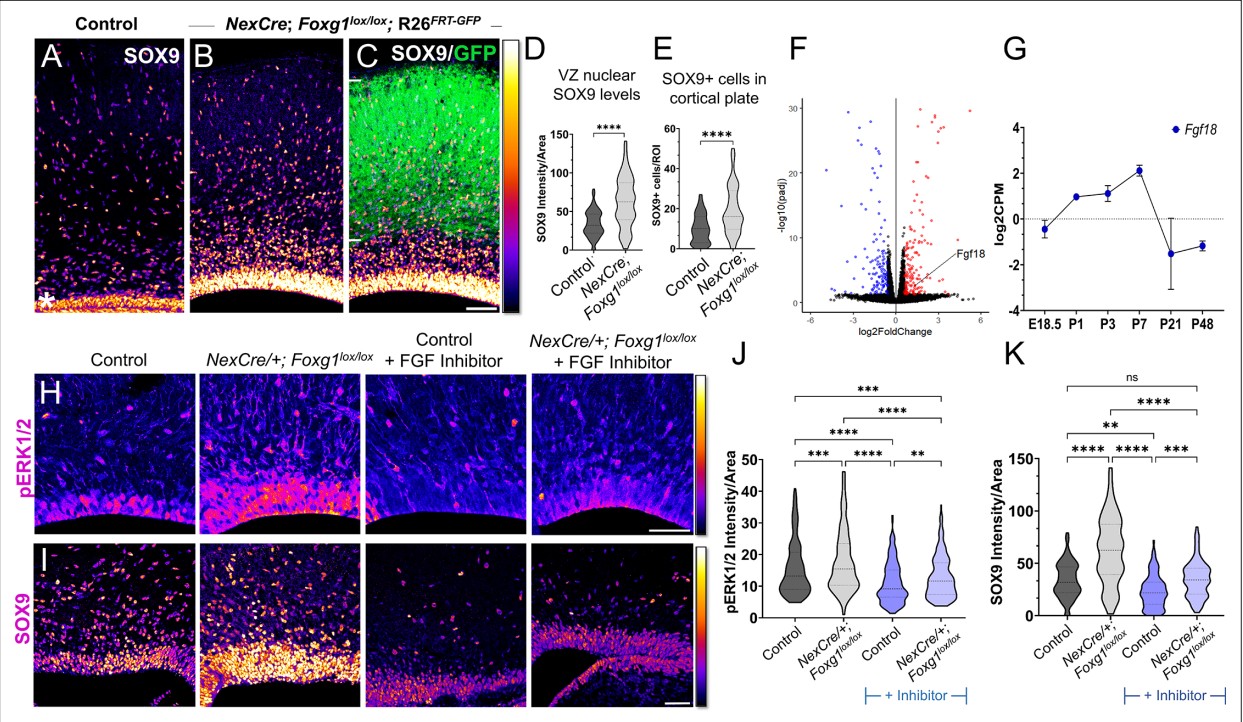

**Figure 5.** Postmitotic neuron *Foxg1* loss-of-function (LOF) leads to premature gliogenesis and upregulation of the MAPK pathway. (**A**) SOX9 staining in the control P0 cortex identifies gliogenic progenitors at the ventricular zone (VZ, white asterisk) and scattered cells throughout the cortical plate. (**B–D**) *NexCre*-driven loss of *Foxg1* is specific to postmitotic neurons, as seen by GFP reporter expression (**C**, white bars) and causes a non-autonomous upregulation of nuclear SOX9 in the VZ progenitors and an increase in the numbers of SOX9+ cells cortical plate (**B**; quantifications: **D**, **E**). (**B**) and (**C**) are images of the same section showing SOX9 alone (**B**) and together with the GFP reporter (**C**). (**F**) Transcriptomic analysis of cortical plate tissue from control and *NexCre/+; Foxg1lox/lox; Rosa26FRT-GFP* reveals a significant upregulation of *Fgf18* upon loss of *Foxg1*. (**G**) *Fgf18* expression in CUX2+ upper-layer cells peaks at P7, as seen in the RNA seq dataset from *Yuan et al., 2022*. (**H**) Examination at E18.5 reveals increased levels of phosphorylated p42/44-ERK1/2 (pERK1/2) within the VZ of *NexCre/+; Foxg1lox/lox* brains, indicative of enhanced FGF signalling. This upregulation of pERK1/2 is reversed upon treatment with the FGF Inhibitor NVP-BGJ398 (**H**; quantifications: **J**). (**I**) In sections from the same brains, levels of SOX9 within the VZ are increased upon postmitotic loss of *Foxg1*, and this is restored to baseline levels upon administration of the inhibitor. (**I**; quantifications: **K**). Quantifications of pERK1/2 in each condition: 233 (Control); 248 (*NexCre/+; Foxg1lox/lox*); 207 (Control+ NVP-BGJ398); 223 (*NexCre/+; Foxg1lox/lox*+NVP-BGJ398) cells from N = 2 brains (biological replicates from two independent experiments). Quantifications of SOX9 levels in each condition: 233 (Control); 234 (*NexCre/+; Foxg1lox/lox*); 228 (Control+ NVP-BGJ398); 205 (*NexCre/+; Foxg1lox/lox*+NVP-BGJ398) cells from N = 3 brains (biological replicates from two independent experiments). Statistical test: Mann–Whitney test (**D, E**); two-way ANOVA with Tukey's correction (**J, K**). *p<0.05, **p<0.01, ***p<0.001, ****p<0.0001. All scale bars: 50 μm.

The online version of this article includes the following figure supplement(s) for figure 5:

**Figure supplement 1.** Postmitotic neuron-specific role of FOXG1 in regulating FGF signalling.

in neurons in the postnatal rodent brain (*Hoshikawa et al., 2002*), displays peak expression in CUX2+ cells in the first postnatal week (*Figure 5—figure supplement 1B*), and is in the same family as FGF8, which is used to induce gliogenesis (*Dinh Duong et al., 2019*). We examined nuclear-phosphorylated p42/44-ERK1/2, indicative of FGF signalling and found it to display increased nuclear localisation specifically in the ventricular zone of *NexCre/+; Foxg1lox/lox* brains (*Figure 5H*). This indicates a net increase in FGF signalling experienced by the progenitors.

To further confirm whether this enhanced gliogenesis upon postmitotic neuron-specific loss of *Foxg1* is indeed via an increase in FGF signalling, we used a pharmacological inhibitor NVP-BGJ398 (infigratinib) that blocks activation of FGFR1/2/3 (*Guagnano et al., 2011*). Dams bearing control and *NexCre/+; Foxg1lox/lox* embryos were administered this inhibitor intraperitoneally from E14.5 to E17.5, and the brains corresponding to each condition were examined at E18.5 for nuclear pERK1/2 levels. In the presence of this inhibitor, nuclear pERK1/2 levels in control VZ cells were reduced compared with untreated embryos, indicating the drug had the predicted effect of abrogating endogenous FGF signalling. Moreover, the increase in nuclear pERK1/2 levels in VZ progenitors in *NexCre/+; Foxg1lox/*

*lox* was no longer seen in the presence of the inhibitor (*Figure 5H and J*). We also examined nuclear SOX9 levels in each condition and found that these paralleled the findings from the FGF inhibitor experiments (*Figure 5I and K*).

Together, these data indicate that FOXG1 regulates signals from postmitotic neurons that non-autonomously trigger a transition to gliogenesis via the FGF-ERK pathway in the progenitors residing in the ventricular zone.

## Attenuation of FGF signalling does not restore neurogenesis in *Foxg1* mutant progenitors but causes premature oligogenesis

Loss of *Foxg1* appears to promote astrogliogenesis via two distinct functions: enhancing FGFR3 expression in progenitors and increasing expression of *Fgf18* in postmitotic neurons. We, therefore,

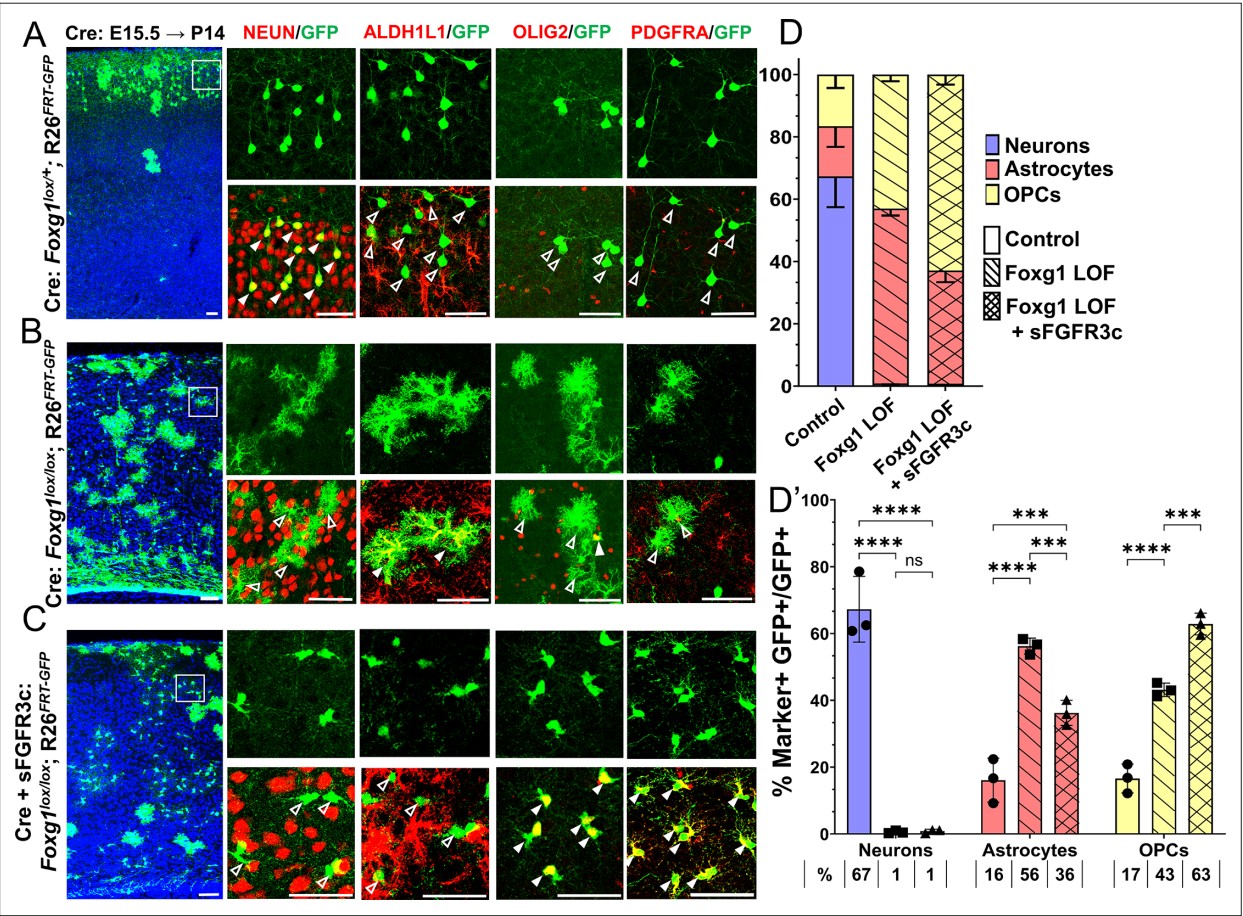

**Figure 6.** *Foxg1-Fgf* double loss-of-function (LOF) leads to premature oligogenesis. (**A–C**) Cre electroporation at E15.5 in control (**A**, *Foxg1^lox/+^*; *Rosa26^FRT-GFP^*) and *Foxg1* LOF (**B**, *Foxg1^lox/lox^*; *Rosa26^FRT-GFP^*) embryos, followed by analysis at P14. GFP+ cells in control brains do not colocalise with ALDH1L1, OLIG2, and PDGFRA staining (**A**), whereas most GFP+ cells in *Foxg1* LOF brains display these markers (**B**). Co-electroporation of *Cre* together with a construct encoding soluble FGFR3c (an FGF-chelator) in *Foxg1^lox/lox^*; *Rosa26^FRT-GFP^* causes a significant increase in the co-localisation of GFP+ cells with oligodendrocyte precursor cells (OPCs) markers such as OLIG2 and PDGFRA (**C**). In each row (**A–C**), the boxes in the leftmost low magnification panels indicate approximate regions from the same section or serial sections shown in the adjacent high magnification panels. A quantitative analysis reveals a drastic reduction of upper-layer neurogenesis at the expense of gliogenesis (astrocytes + OPCs) upon loss of *Foxg1* and an additional increase in the percentage of OPCs with the additional abrogation of FGF signalling (**D, D'**). n = 4069 (Control), 3970 (*Foxg1* LOF), 3332 (*Foxg1* LOF+sFgfr3c) from N = 3 brains (biologically independent replicates). Statistical test: two-way ANOVA.with Tukey's correction *p<0.05, **p<0.01, ***p<0.001, ****p<0.0001. All scale bars: 50 µm.

The online version of this article includes the following figure supplement(s) for figure 6:

**Figure supplement 1.** *sFgfr3c* overexpression at E15.5 in *Rosa26^FRT-GFP^* background leads to prolonged neurogenesis.

**Figure supplement 2.** FGF signalling is modulated upon induction of sFGFR3c in E15.5 progenitors.

**Figure supplement 3.** *Foxg1* overexpression at E15.5 in *Rosa26^FRT-GFP^* background leads to prolonged neurogenesis.

tested whether reducing the available FGF ligands may be sufficient to restore neurogenesis in *Foxg1* LOF progenitors. We used a construct encoding soluble FGFR3 chelator (sFgfr3c), which has been effectively used by other studies to sequester and limit the availability of FGFs using in utero electroporation (*Dinh Duong et al., 2019*; *Fukuchi-Shimogori and Grove, 2001*).

As before (*Figure 1*), embryos were electroporated at E15.5, and the brains were harvested at P14 and examined for glial markers either common to glial precursor cells (OLIG2) or exclusive to either OPCs (PDGFRA) or astrocytes (ALDH1L1). For two of the conditions, control (Cre electroporation in *Foxg1*<sup>lox/+</sup>) and *Foxg1* LOF alone (Cre electroporation in *Foxg1*<sup>lox/lox</sup>), brains from the same set of experiments presented in *Figure 1* were used from which additional sections were examined for OLIG2 and PDGFRA (*Figure 6A and B*). In controls, the majority of GFP+ cells were neurons that did not express any glial marker (*Figures 1, 6A and D*). Neurogenesis was significantly reduced upon loss of *Foxg1* alone, and there was a corresponding increase in both types of glia (*Figure 6B and D*). For the new experimental condition, which involves a combined loss of *Foxg1* and abrogation of FGF signalling, we co-electroporated *Cre* with sFgfr3c into *Foxg1*<sup>lox/lox</sup> embryos. Strikingly, this condition did not restore neurogenesis but instead shifted the cell fate towards OLIG2+, PDGFRA+, and ALDH1L1⁻ OPCs (*Figure 6C and D*) Conversely, neurogenesis was enhanced at the expense of gliogenesis when *Cre* and sFgfr3c were electroporated in *Foxg1*<sup>lox/+</sup>; *Rosa26*<sup>FRT-GFP</sup> (Control) embryos (*Figure 6—figure supplement 1*), consistent with previous reports of sFgfr3c overexpression in wild-type brains (*Dinh Duong et al., 2019*). To verify the functional consequences of FGF signalling, we quantified nuclear phospho-ERK1/2 levels in the different conditions 1 day after electroporation, when the process of cell fate decision would be underway in progenitors. Loss of *Foxg1* at E15.5 led to an increase in nuclear localisation of phospho-ERK1/2 by E16.5, and this effect was abrogated by co-electroporation of sFgfr3c (*Figure 6—figure supplement 2*).

These results indicate that upper-layer neurogenesis can neither proceed normally nor be restored by abrogating FGF signalling in the absence of FOXG1. Instead, the progenitors display a premature progression to an OPC fate, which appears to be a permissive cell fate in the context of the combined loss/decrease of FOXG1 and FGF signalling, respectively.

We tested the effects of prolonged *Foxg1* expression on progenitor gliogenic potential. We performed *Foxg1* overexpression using the *Rosa26*<sup>FRT-GFP</sup> background to examine the lineage arising from E15.5 progenitors scored at P14. We co-electroporated a *Foxg1* full-length construct together with *Cre* in control *Foxg1*<sup>lox/+</sup>; *Rosa26*<sup>FRT-GFP</sup> embryos at E15.5. This overexpression led to an increase in the neurons produced by E15.5 progenitors to 98% compared with 67% in controls (*Figure 6—figure supplement 3*). This suggests that a prolonged *Foxg1* expression extends the ability of progenitors to produce neurons at the expense of glia. This is consistent with the established role of FOXG1 as a neurogenic factor (*Falcone et al., 2019*).

In summary, the results uncover a fundamental role of FOXG1 in regulating the transition of cortical progenitors from neurogenesis to gliogenesis and also highlight that upper-layer neurogenesis is not possible in the absence of this critical factor.

## Discussion

Cell-type diversity arises due to genetic mechanisms that regulate how progenitors give rise to cells with unique identities. In the nervous system, neurons, astrocytes, and oligodendrocytes have distinct functions, and the production of the correct numbers of each of these three cell types must be carefully controlled. How these cells arise from common progenitors is one of the outstanding questions in developmental neuroscience. Progenitors regulate the cell fate of their progeny via cell-autonomous, that is, intrinsic programs, and cell non-autonomous, that is, extrinsic signalling mechanisms. Understanding how these regulatory processes are orchestrated and how the progenitor integrates these effects is crucial in elucidating how these transitions in cell fate arise. Our study reveals two novel functions of transcription factor FOXG1 in controlling both the expression of FGF family ligands in cortical neurons, as well as the sensitivity of progenitors to these FGF ligands via the expression of the receptor FGFR3. Thus, FOXG1 integrates the production and receptivity to extrinsic signals so progenitors may unfold cell-intrinsic programs to produce glia (*Figure 7*).

Whereas the different stages of neurogenesis have been well studied (*Huilgol et al., 2023*; *Mukhtar and Taylor, 2018*), the progression from neurogenesis to gliogenesis and through the various stages of gliogenesis has been less well understood. *Foxg1* was previously identified as a regulator of

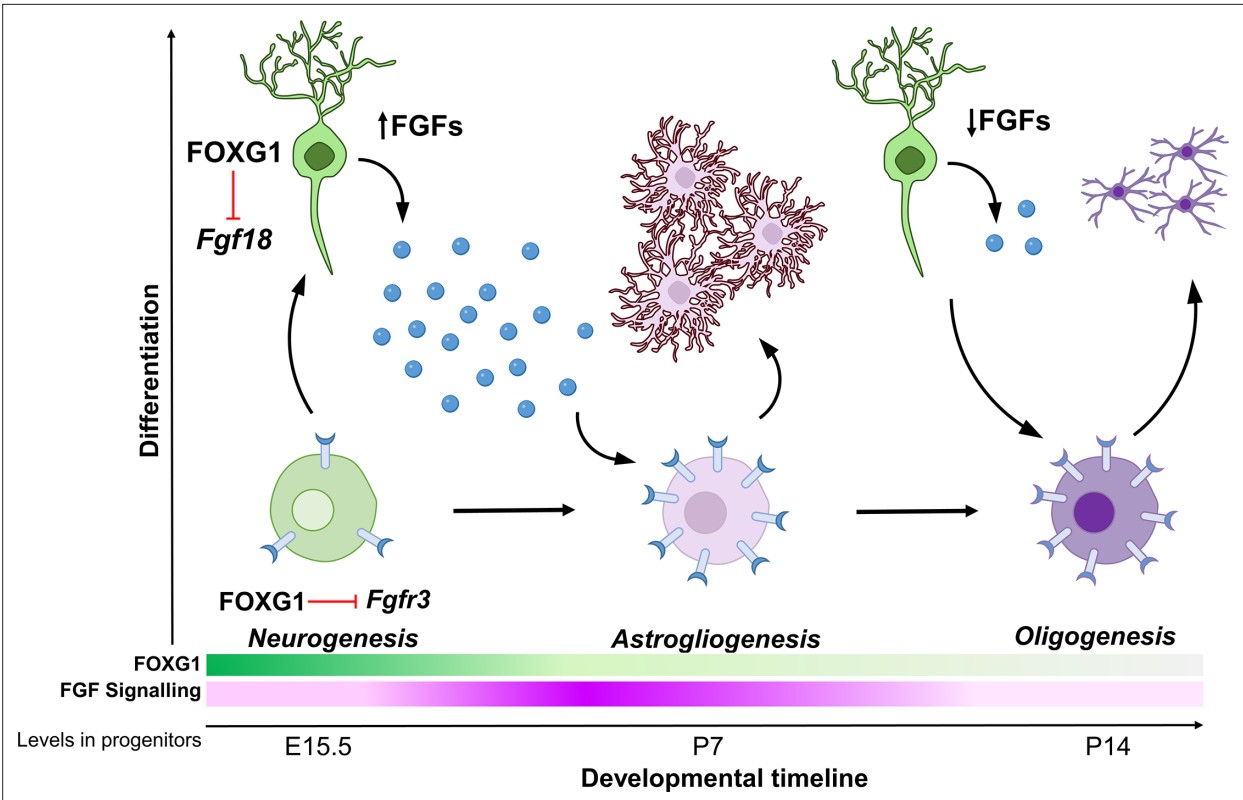

**Figure 7.** Schematic depicting the regulation of gliogenesis by FOXG1. In neurogenic progenitors, FGFR3 levels are suppressed by FOXG1. As time progresses, FOXG1 levels decrease within progenitors (expression gradient in green), and FGFR3 levels increase (expression gradient in magenta), making progenitors more sensitive to FGF signalling. Concomitantly, postmitotic neurons secrete factors, including FGF18, which is also under FOXG1 regulation. FGF signalling drives progenitors towards astrogliogenesis. Later, when both, FOXG1 and FGF levels are low, the progenitors transition to oligogenesis.

sequential neurogenesis in the neocortex via controlling the timing and order of generation of specific neuronal subtypes (*Toma et al., 2014*; *Hou et al., 2020*; *Liu et al., 2022*; *Cargnin et al., 2018*), but its endogenous role in regulating gliogenesis was not examined. The suggestion that FOXG1 may regulate the neuron-glia cell fate switch came from *Foxg1* overexpression in E12 cortical progenitors in vitro, which caused decreased gliogenesis (*Falcone et al., 2019*; *Frisari et al., 2022*). However, in these studies, *Foxg1* knockdown did not result in gliogenesis. Therefore, the endogenous role of FOXG1 could not be established (*Falcone et al., 2019*; *Frisari et al., 2022*).

We discovered a novel link between FOXG1 and FGF signalling. Cortical progenitors express *Fgfr1*, *r2*, and *r3* from early stages (*Dinh Duong et al., 2019*; *Bansal et al., 2003a*), and the expression levels increase during gliogenesis (*Dinh Duong et al., 2019*). Our work demonstrates that the timing of the cessation of neurogenesis and the subsequent emergence of glial lineages is controlled by FOXG1 in cortical progenitors in vivo. Furthermore, we identify a key target, *Fgfr3*, via which FOXG1 controls the response of the progenitor to extrinsic FGF signalling. Importantly, we find that progenitors at E14.5 as well as E15.5 produce glia upon loss of *Foxg1*. This indicates that *Foxg1* mutant progenitors do not simply accelerate to the immediate next stage of cell-type production, that is, layer II/III instead of layer 4, but undergo a switch to producing a different class, astroglia. A decrease in FOXG1 in apical progenitors (*Figure 1—figure supplement 1*; *Falcone et al., 2019*) may be an endogenous mechanism for the transition of upper-layer neurogenesis to gliogenesis in progenitors. Intriguingly, E13.5 progenitors do not progress to premature gliogenesis upon loss of *Foxg1*, consistent with the idea that progenitors transition through distinct competent states (*Telley et al., 2016*), an avenue that motivates future directions of inquiry.

Postmitotic neurons are known to provide cues to progenitors to initiate the production of the next wave of cells (*Seuntjens et al., 2009*; *Barnabé-Heider et al., 2005*; *Hoshikawa et al., 2002*). In this context, we identify an additional function of FOXG1 in regulating the expression of FGF ligands

from postmitotic neurons that results in a net increase in FGF signalling experienced by progenitors. This provides an elegant mechanism for timing the onset of astrogliogenesis after upper-layer neurogenesis is complete. In vivo, *Foxg1* expression is dynamically downregulated in pyramidal neuronal precursors as they migrate in the intermediate zone (*Miyoshi and Fishell, 2012*), which provides a potential endogenous mechanism for the modulation of FGFs secreted by postmitotic neurons. Further investigation is needed to determine whether the regulation of *Fgf18* by FOXG1 occurs through direct transcriptional control or indirect mechanisms. Additional regulatory mechanisms within maturing postmitotic neurons that remain unidentified may also participate in such feedback signalling to progenitors.

FGF signalling has essential roles in the early patterning of the cortical primordium *Fukuchi-Shimogori and Grove, 2001*; *Fukuchi-Shimogori and Grove, 2003*; *Crossley et al., 2001* and the guidance of major axon tracts in the brain (*Tole et al., 2006*). However, its importance during late embryonic and postnatal stages has yet to be fully explored. Recent evidence suggests that FGF signalling regulates neurogenesis and neuronal migration (*Dinh Duong et al., 2019*; *Kon et al., 2019*; *Szczurkowska et al., 2016*). Depletion of FGF from E15.5 progenitors leads to prolonged neurogenesis in the cortex (*Dinh Duong et al., 2019*). However, reducing FGF signalling in *Foxg1* mutant progenitors does not restore neurogenesis but causes a premature production of OPCs. These data indicate that FOXG1 is essential for neuronal production, and in its absence, the removal of FGF is insufficient to prolong the neurogenic programme in progenitors. Overexpression of *Foxg1*, however, does restore neurogenesis despite augmented FGF signalling, and overexpression of *Foxg1* dramatically enhances the neurogenic potential of the progenitor at the expense of gliogenesis. This establishes that FOXG1 is necessary and sufficient for neuronal fate regardless of FGF signalling levels.

A rich body of work has explored how FGF, SHH, BMP, and WNT signalling regulate the proliferation and differentiation of OPCs in vitro and in the spinal cord (*McCaughey-Chapman and Connor, 2023*; *Manzari-Tavakoli et al., 2022*; *El Waly et al., 2014*; *Farreny et al., 2017*). However, there is a lack of information on how OPCs arise from gliogenic progenitors and whether there are differences in gliogenic mechanisms between the spinal cord and cerebral cortex, given that the initial molecular context of progenitors in these two structures is vastly different (*Zeisel et al., 2018*; *Silbereis et al., 2016*).

Our data indicates that once a neocortical progenitor can no longer generate neurons due to loss of *Foxg1*, it appears to be forced into a gliogenic mode, producing astrocytes if FGF signalling is present and OPCs if FGF signalling is abrogated. Taking the results from *Dinh Duong et al., 2019* and our study together, the emerging interpretation is that for FOXG1 mutant progenitors, FGF signalling seems to be critical for astrocyte production, but it does not appear to be required for oligogenesis. The enhanced oligogenesis we report may be the result of progenitors being unable to either return to neurogenesis due to the absence of *Foxg1* or to maintain astrogliogenesis due to a reduction of FGF signalling. In vivo, a decline in FGFs available to progenitors may initiate OPC production. This is consistent with a general mechanism in which the regulated secretion of the ligands from previously born cells governs the timing of generation and proliferation of later-born cells (*Seuntjens et al., 2009*). In the particular context of astrogliogenesis and oligogenesis, a more in-depth analysis of the affinity of FGF-FGFR binding in progenitors and specified glial cells is required to understand the fine-tuning of these processes (*Bansal et al., 1996*; *Bansal et al., 2003b*).

FOXG1 modulates the cell-intrinsic response of progenitors to cell-extrinsic FGF signalling, thereby regulating the production of neurons, astroglia, and oligodendrocytes, the three major components of brain circuitry. In parallel, FOXG1 also regulates FGF ligand expression in postmitotic neurons, which triggers gliogenesis in progenitors. In humans, either gain- or loss-of-function *Foxg1* mutations cause 'FOXG1 syndrome', an autism spectrum disorder (*Hou et al., 2020*; *Hettige and Ernst, 2019*; *Florian et al., 2012*) that results in a range of conditions that include microcephaly, seizures and may also increase the propensity for glioblastoma or myelination deficits (*Hou et al., 2020*). Our findings offer mechanistic insights into potentially novel deficits that may underlie aspects of Foxg1 syndrome-associated dysfunction.

# Materials and methods

**Key resources table**

| Reagent type (species) or resource | Designation | Source or reference | Identifiers | Additional information |
|---|---|---|---|---|
| Strain, strain background (*Mus musculus*) | *Foxg1*$^{lox/lox}$; *Rosa26*$^{FRT-GFP}$ | PMID:22726835 | | |
| Strain, strain background (*M. musculus*) | MADM-12 GT/GT | PMID:34161767 | | |
| Strain, strain background (*M. musculus*) | MADM-12 TG/TG | PMID:34161767 | | |
| Strain, strain background (*M. musculus*) | hGFAP-CreERT2 | Jackson Laboratory | Strain No.: 012849 | |
| Strain, strain background (*M. musculus*) | Ai9 | Jackson Laboratory | Strain No.: 007909 | |
| Strain, strain background (*M. musculus*) | NexCre/+ (*Neurod6*-Cre) | PMID:17146780 | | |
| Transfected construct (*M. musculus*) | pCAGG-IRES-eGFP | PMID:22726835 | | Gift from Prof. Gord Fishell |
| Transfected construct (*M. musculus*) | pCAGG-IRES-FOXG1-EGFP | PMID:22726835 | | Gift from Prof. Gord Fishell |
| Transfected construct (*M. musculus*) | pCAGG-FGF8 | PMID:31175212 | | Gift from Prof. Hiroshi Kawasaki |
| Transfected construct (*M. musculus*) | pCAGG-sFGFR3c | PMID:31175212 | | Gift from Prof. Hiroshi Kawasaki |
| Antibody | Biotinylated GFP (goat polyclonal) | Abcam | Catalog number: ab6658 | 1:200 |
| Antibody | NEUN (rabbit monoclonal) | Thermo Fisher Scientific | Catalog number: 702022 | 1:200 |
| Antibody | ALDH1L1 (rabbit polyclonal) | Abcam | Catalog number: ab87117 | 1:200 |
| Antibody | OLIG2 (rabbit polyclonal) | Merck Millipore | Catalog number: AB9610 | 1:200 |
| Antibody | SOX9 (rabbit monoclonal) | Abcam | Catalog number: ab185230 | 1:200 |
| Antibody | KI67 (rabbit monoclonal) | Thermo Fisher Scientific | Catalog number: MA5-14520 | 1:1000 |
| Antibody | RFP (mouse monoclonal) | Thermo Fisher Scientific | Catalog number: MA5-15257 | 1:200 |
| Antibody | Phospho p42/44 MAPK (rabbit monoclonal) | Cell Signaling Technology | Catalog number: 4370S | 1:200 |
| Antibody | FGFR3 (rabbit polyclonal) | Affinity Biosciences | Catalog number: AF0160 | 1:100 |
| Antibody | NF1A (rabbit polyclonal) | Abcam | Catalog number: ab228897 | 1:500 |
| Antibody | PAX6 (mouse monoclonal) | Thermo Fisher Scientific | Catalog number: MA1-109 | 1:500 |
| Antibody | EOMES (rat monoclonal) | Thermo Fisher Scientific | Catalog number: 14-4875-82 | 1:200 |
| Antibody | SOX2 (mouse monoclonal) | Thermo Fisher Scientific | Catalog number: MA1-014 | 1:200 |
| Antibody | CD140a (mouse monoclonal) | BD Biosciences | Catalog number: 558774 | 1:500 |

*Continued on next page*

*Continued*

| Reagent type (species) or resource | Designation | Source or reference | Identifiers | Additional information |
|---|---|---|---|---|
| Antibody | FOXG1 (rabbit polyclonal) | TakaraBio | Catalog number: M227 | 1:200 |
| Antibody | Anti-rabbit 568 (goat polyclonal) | Thermo Fisher Scientific | Catalog number: A11011 | 1:200 |
| Antibody | Streptavidin Alexa 488 Conjugate Dye | Thermo Fisher Scientific | Catalog number: S32354 | 1:200 |
| Antibody | Anti-rabbit 647 (donkey polyclonal) | Thermo Fisher Scientific | Catalog number: A31573 | 1:200 |
| Antibody | Anti-mouse 568 (goat polyclonal) | Thermo Fisher Scientific | Catalog number: A11004 | 1:200 |
| Antibody | Anti-rat 647 (goat polyclonal) | Thermo Fisher Scientific | Catalog number: A21247 | 1:200 |
| Antibody | Anti-rat 568 (goat polyclonal) | Thermo Fisher Scientific | Catalog number: A11077 | 1:200 |
| Antibody | Anti-rabbit 488 (goat polyclonal) | Thermo Fisher Scientific | Catalog number: A11034 | 1:200 |
| Commercial assay or kit | FlashTag | Thermo Fisher Scientific | SKU C34554 | |
| Commercial assay or kit | HBSS without calcium/magnesium | Thermo Fisher Scientific | Catalog number: 14170112 | |
| Commercial assay or kit | HBSS with calcium/magnesium | Thermo Fisher Scientific | Catalog number: 14025092 | |
| Commercial assay or kit | 0.25% Trypsin | Thermo Fisher Scientific Gibco | Catalog number: 15400054 | |
| Chemical compound, drug | Corn oil | Sigma-Aldrich | Catalog number: 8267 | |
| Chemical compound, drug | Triton-X100 | Sigma-Aldrich | CAS No.: 9036-19-5 | |
| Chemical compound, drug | Tamoxifen | Sigma-Aldrich | Catalog number:T5648 | |
| Chemical compound, drug | NVP-BGJ398 | MedChemExpress | Catalog number: HY-13311 | |
| Software, algorithm | FastQC | Babraham Bioinformatics | Other | https://www.bioinformatics.babraham.ac.uk/projects/fastqc/ |
| Software, algorithm | HISAT2 | PMID:31375807 | | |
| Software, algorithm | DESeq2 | PMID:25516281 | | |
| Software, algorithm | Shiny GO | PMID:31882993 | | |
| Software, algorithm | SRA Toolkit | SRA Toolkit Development Team | Other | https://trace.ncbi.nlm.nih.gov/Traces/sra/sra.cgi?view=software |
| Software, algorithm | Bowtie2 | PMID:22388286 | | |
| Software, algorithm | Homer | PMID:20513432 | | |
| Software, algorithm | BEDTools | PMID:20110278 | | |
| Software, algorithm | IGV | PMID:21221095 | | |
| Software, algorithm | Fiji | PMID:22743772 | | |

## Mice

All procedures followed the Tata Institute of Fundamental Research Animal Ethics Committee (TIFR-IAEC) guidelines (IAEC approval no.: TIFR/IAEC/2022-3).

The *Foxg1^lox/lox^; Rosa26^FRT-GFP^* mouse line used in this study is described in *Miyoshi and Fishell, 2012*. MADM-12 GT/GT and MADM-12 TG/TG lines are described in *Contreras et al., 2021*. hGFAP-CreERT2 (Strain #:012849) and *Ai9* reporter mouse line (Strain #:007909) were obtained from Jackson Laboratory. The *NexCre/+* mouse line was obtained from Klaus Nave, Max Planck Institute for Experimental Medicine (*Goebbels et al., 2006*).

All animals were kept at an ambient temperature and humidity, with a 12-hour light-dark cycle and food available ad libitum. Noon of the day of the vaginal plug was designated as embryonic day 0.5 (E0.5). Both male and female animals were used for all experiments.

Primers for genotyping were (expected band sizes):

> Foxg1cKO F: CCACTCCGAACCCGCTGG,
> Foxg1cKO R: AGGCTGTTGATGCTGAACGA, (mutant: 190 bp, WT: 156 bp);
> FRT Reporter: RCE-Rosa1: CCCAAAGTCGCTCTGAGTTGTTATC,
> RCE-Rosa2: GAAGGAGCGGGAGAAATGGATATG,
> RCE-Cag3: CCAGGCGGGCCATTTACCGTAAG, (WT: 550 bp, FRT: 350 bp).
> MADM 12 Cassette:
> Chr 12 WT F: CACTAAGCTCCACTCGCACC,
> Chr 12 WT R: TCCCTCATGATGTATCCCCT,
> MADM R: TCA ATG GGC GGG GGT CGT T, (WT: 322 bp, MADM Cassette: 200 bp).
> Cre F: ATTTGCCTGCATTACCGGTC.
> Cre R: ATCAACGTTTTCTTTTCGG (Cre: 350 bp)
> NexCre Forward: GAGTCCTGGAATCAGTCTTTTTC
> NexCre Reverse: AGAATGTGGAGTAGGGTGAC
> NexCre Mutant Reverse: CCGCATAACCAGTGAAACAG (WT: 770 bp, NexCre: 525 bp)

## In utero electroporation

In utero electroporation was performed as previously described (*Pal et al., 2021*). Embryos were injected with 1–2 µL of plasmid DNA solution dissolved in nuclease-free water with 0.1% fast green with plasmid DNA into the lateral ventricle through the uterine wall using a fine glass microcapillary (Sutter capillaries #B100-75-10). Constructs pCAGG-IRES-eGFP and pCAGG-IRES-FOXG1-EGFP were gifts from Gord Fishell, Harvard Medical School. pCAGG-FGF8 and pCAGG-sFGFR3c were gifts from Hiroshi Kawasaki, Kanazawa University.

## Tissue preparation

Embryos were isolated in ice-cold PBS. Embryonic brains were dissected and fixed overnight in 4% (wt/vol) paraformaldehyde at 4°C overnight and then cryoprotected by transferring to 30% (wt/vol) sucrose-PBS until sectioning. Postnatal mice were anaesthetised using thiopentone and transcardially perfused with 4% (wt/vol) paraformaldehyde in phosphate buffer, followed by overnight fixation and then cryoprotected by transferring to 30% sucrose-PBS until sectioning. The brains were sectioned at 30 µm and 40 µm for MADM using a freezing microtome (Leica SM2000R).

## FGF inhibitor administration and analysis

We used the FGFR inhibitor NVP-BGJ398, a selective inhibitor of FGF receptors 1, 2, and 3 for blocking FGF signalling (*Guagnano et al., 2011*). Pregnant dams were administered NVP-BGJ398 (10 mg/kg body weight in DMSO and corn oil; MedChemExpress, Catalog number: HY-13311) or vehicle solution (corn oil) by intraperitoneal injection twice a day from E14.5 to E17.5, eight treatments in total.

## Immunohistochemistry

Brains were sectioned (30 µm), mounted on Superfrost plus glass microscope slides (Catalog number: 71869-10), and dried for 2 hours at room temperature (RT). Three washes were given for 5 minutes each (3 * 5′) in phosphate buffer. All antibodies except FGFR3 and PDGFRα required antigen retrieval at 90–95°C in 10 mM sodium citrate buffer (pH 6.0) for 10 minutes. Sections were immersed in blocking solution (5% [vol/vol] horse serum in phosphate buffer with 0.1% [vol/vol] Triton X-100 [Sigma; X100]) for 1 hour at RT. Incubation with primary antibody was performed in phosphate buffer containing 0.1% (vol/vol) Triton X-100 and 2.5% (vol/vol) horse serum at 4°C overnight. For postnatal brain sections,

free-floating immunohistochemistry was performed. Sections were given three washes for 5 minutes each (3 * 5′) in phosphate buffer and then permeabilised with phosphate buffer containing 0.3% (vol/vol) Triton X-100 for 10 minutes. Blocking was done with 5% (vol/vol) horse serum in phosphate buffer with 0.3% (vol/vol) Triton X-100 for 1 hour at RT. This was followed by primary antibody treatment in phosphate buffer containing 0.3% (vol/vol) Triton X-100 and 2.5% (vol/vol) horse serum overnight at 4°C. The sections were washed in phosphate buffer, followed by the appropriate secondary antibody for 2 hours at RT.

This was followed by three washes for 5 minutes each (3 * 5′) in phosphate buffer and DAPI staining for 10 minutes, after which the sections were washed with phosphate buffer for 15 minutes (3 * 5′). The slides were then mounted with Fluoroshield (Sigma Catalog number: F6057 or F6182). Please refer to the Key Resources Table for detailed information on antibodies.

## Image acquisition and analysis

Fluorescence images were taken using an Olympus FluoView 3000 confocal microscope with FluoView software. All the image analysis was done on Fiji-ImageJ. A nonlinear operation such as gamma correction was not performed in the figures. Brightness and contrast adjustments were performed identically for control and mutant conditions. Cell counting was performed using the Cell Counter plugin in Fiji.

Intensity quantifications for Phospho-p42/44-ERK1/2 and SOX9 were done by imaging Control and mutant samples at the same laser, gain and offset settings on the Olympus FV3000 confocal microscope. Images were stacked based on one-cell thickness stacks, that is, approximately 5 µm. Intensity quantification was done by drawing regions of interest (ROIs) around the nuclei using DAPI or SOX2, followed by intensity quantification of Phospho-p42/44-ERK1/2 or SOX9 and calculating mean intensity per unit ROI.

## Histone and FOXG1 ChIP-seq analysis

FASTQ files deposited by previously published manuscripts (*Liu et al., 2017*; *Cargnin et al., 2018*) were obtained using SRA Toolkit's fastq-dump command and aligned to mouse reference genome mm10 using Bowtie2 (*Langmead and Salzberg, 2012*). Peak calling and annotation were performed using Homer using default options and command style "-histone" for the H3K27Me3 and H3K4Me3 datasets. For the FOXG1 ChIP-seq analysis, peaks were called using all "-style" options and only the top 50% peaks called were considered for further analysis for all the ChIP-seq datasets. Peaks were intersected using BEDTools software *Quinlan and Hall, 2010* and annotated using Homer (*Heinz et al., 2010*) annotatePeaks.pl function, and genome view plots were created using IGV (*Robinson et al., 2011*).

## Fluorescence-assisted cell sorting

*Foxg1* mutant cells were obtained by administering tamoxifen (Catalog number: T5648) prepared in corn oil (Sigma; Catalog number: 8267) to E15.5 hGFAP-CreERT2; *Foxg1^{lox/lox}*; Ai9 dams, at 40 µg/g body weight dose and FACS-isolated using the Ai9 reporter at E17.5. We used FlashTag-labelled (Thermo Fisher Scientific SKU C34554) progenitors at E15.5 for controls and collected the cells at E17.5. Cortical tissue was dissected in HBSS without calcium/magnesium (Thermo Fisher Scientific Catalogue number: 14170112), and a single-cell suspension was prepared in HBSS with calcium/magnesium (Thermo Fisher Scientific Catalogue number: 14025092) using 0.25% Trysin (Thermo Fisher Scientific Gibco Catalogue number: 15400054) and 70 um cell strainer. FACS was performed using BD Aria Fusion (BD Biosciences) with the 568 and 488 lasers using an 85 µm nozzle. Singlets were selected using forward scatter and side scatter. Cells were selected for collection based on their RFP signal (Mutant) or GFP signal (Control).

## RNA seq: Sample preparation and analysis

Cells obtained from FACS were stored in RNAlater until extraction. 400,000 cells were pooled from at least two brains to obtain a biological replicate. RNA extraction and sequencing were performed on three control replicates and four *Foxg1* mutant replicate cell suspensions. 1 ug of RNA was used to obtain the cDNA library, and sequencing was performed on an Illumina platform to achieve 150 bp reads to generate 30 million paired-end reads per sample. Fastq QC was performed as described in https://www.bioinformatics.babraham.ac.uk/projects/fastqc/ and reads >30 Phred scores were

aligned using HISAT2 (*Kim et al., 2019*). Feature counts were used to quantify the number of reads per transcript. Differential expression analysis was performed using DESeq2 (*Love et al., 2014*). A log2FoldChange cutoff of 0.58 and p-value <0.05 was used to identify DEGs. GO analysis was performed using Shiny GO (*Ge et al., 2020*). Genes described as Astrocyte-enriched were obtained from *Telley et al., 2019*.

## Acknowledgements

We thank the animal house staff of the Tata Institute of Fundamental Research, Mumbai (TIFR), for their excellent support; Gordon Fishell (Harvard Medical School, USA), and Goichi Miyoshi (Gunma University, Japan) for the *Foxg1* floxed mouse line; Hiroshi Kawasaki (Kanazawa University, Japan) for the plasmids pCAG-FGF8 and pCAG-sFgfr3c; Soo Kyung Lee (University at Buffalo, The State University of New York, USA) for the *Foxg1*^lox/lox genotyping primers and protocol. We thank Deepak Modi and Vainav Patel (National Institute for Research in Reproductive and Child Health, NIRRCH, Mumbai, India) for the use of the NIRRCH FACS Facility, and the staff of the NIRRCH and TIFR FACS facilities for their assistance. We thank Denis Jabaudon (University of Geneva, Switzerland) for his critical comments on the manuscript and members of the Jabaudon lab for helpful discussions. This work was funded by the Department of Atomic Energy (DAE), Govt. of India (Project Identification no. RTI4003, DAE OM no. 1303/2/2019/R&D-II/DAE/2079).

## Additional information

### Funding

| Funder | Grant reference number | Author |
| --- | --- | --- |
| Department of Atomic Energy, Government of India | Project Identification no. RTI4003, DAE OM no. 1303/2/2019/R&D-II/DAE/2079 | Mahima Bose<br>Ishita Talwar<br>Varun Suresh<br>Urvi Mishra<br>Shiona Biswas<br>Anuradha Yadav<br>Shital T Suryavanshi<br>Shubha Tole |

The funders had no role in study design, data collection and interpretation, or the decision to submit the work for publication.

### Author contributions

Mahima Bose, Conceptualization, Data curation, Formal analysis, Validation, Visualization, Methodology, Writing – original draft, Writing – review and editing; Ishita Talwar, Conceptualization, Data curation, Formal analysis, Validation, Visualization; Varun Suresh, Conceptualization, Data curation, Formal analysis, Validation, Visualization, Methodology; Urvi Mishra, Data curation, Formal analysis, Validation, Visualization, Methodology; Shiona Biswas, Data curation, Formal analysis, Visualization; Anuradha Yadav, Data curation; Shital T Suryavanshi, Methodology; Simon Hippenmeyer, Resources; Shubha Tole, Conceptualization, Supervision, Funding acquisition, Investigation, Project administration, Writing – review and editing

### Author ORCIDs

Mahima Bose ![ORCID] https://orcid.org/0000-0002-9069-7889
Varun Suresh ![ORCID] https://orcid.org/0000-0002-9177-9843
Urvi Mishra ![ORCID] https://orcid.org/0000-0002-2299-7417
Shiona Biswas ![ORCID] http://orcid.org/0000-0003-4308-2133
Simon Hippenmeyer ![ORCID] https://orcid.org/0000-0003-2279-1061
Shubha Tole ![ORCID] https://orcid.org/0000-0001-6584-443X

### Ethics

All procedures followed the Tata Institute of Fundamental Research Animal Ethics Committee (TIFR-IAEC) guidelines. (IAEC approval no.: TIFR/IAEC/2022-3).

Reviewer #1 (Public review): https://doi.org/10.7554/eLife.101851.3.sa1
Reviewer #2 (Public review): https://doi.org/10.7554/eLife.101851.3.sa2
Author response https://doi.org/10.7554/eLife.101851.3.sa3

## Additional files

### Supplementary files

Supplementary file 1. FOXG1 occupied and differentially expressed genes in progenitors. Dataset 1 (separate file): Sheet 1: differentially expressed genes identified using DESeq2 in the *Foxg1* LOF cells vs. Control at E17.5 related to *Figure 3A*. Sheet 2: list intersecting regions between the FOXG1 ChIP-seq and bivalent marks H3K27me3 and H3K4me3.

Supplementary file 2. Neuron-specific *Foxg1* loss induced differentially expressed secreted factors in the cortical plate. Dataset 2 (separate file): Sheet 1: differentially expressed genes identified using DESeq2 in *NexCre/+; Foxg1* LOF vs Control cortical plate cells related to *Figure 5E*. Sheet 2: list of differentially expressed secreted molecules subset from the *NexCre/+; Foxg1* LOF vs Control RNA seq dataset.

MDAR checklist

### Data availability

All RNAseq samples generated for this manuscript are deposited under PRJNA886320 and GSE253919.

The following dataset was generated:

| Author(s) | Year | Dataset title | Dataset URL | Database and Identifier |
|---|---|---|---|---|
| Bose M, Suresh V, Talwar I, Mishra U, Tole S | 2025 | Dual role of FOXG1 in regulating gliogenesis in the developing neocortex via the FGF signalling pathway | https://www.ncbi.nlm.nih.gov/geo/query/acc.cgi?&acc=GSE253919 | NCBI Gene Expression Omnibus, GSE253919 |

The following previously published datasets were used:

| Author(s) | Year | Dataset title | Dataset URL | Database and Identifier |
|---|---|---|---|---|
| Telley L, Agirman G, Prados J, Amberg N, Fièvre S, Oberst P, Bartolini G, Vitali I, Cadilhac C, Hippenmeyer S, Nguyen L, Dayer A, Jabaudon D | 2019 | Temporal patterning of apical progenitors and their daughter neurons in the developing neocortex | https://www.ncbi.nlm.nih.gov/geo/query/acc.cgi?acc=GSE118953 | NCBI Gene Expression Omnibus, GSE118953 |
| Liu J, Wu X, Zhang H, Pfeifer GP, Lu Q | 2017 | Dynamics of RNA polymerase II pausing and bivalent histone H3 methylation during neuronal differentiation in brain development | https://www.ncbi.nlm.nih.gov/geo/query/acc.cgi?acc=GSE93011 | NCBI Gene Expression Omnibus, GSE93011 |
| Cargnin F, Kwon J-S, Katzman S, Chen B, Lee JW, Lee S-K | 2018 | ChIP-seq for Foxg1 in E14-15 cortex | https://www.ncbi.nlm.nih.gov/geo/query/acc.cgi?acc=GSE96070 | NCBI Gene Expression Omnibus, GSE96070 |

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
