## [Editor Report · eLife Assessment]

This **important** study provides **convincing** evidence that developing neurons in the neocortex regulate glial cell development. The data demonstrates that the transcription factor FOXG1 negatively regulates gliogenesis by controlling the expression of a member of the FGF ligand family and by suppressing the receptor for this ligand in developing neurons. This study leads to a new understanding of the cascade of events regulating the timing of glial development in the neocortex.

---

## [Referee Report · Reviewer #1 (Public review)]

Summary:

In this paper, Bose et al. investigated the role of Foxg1 transcription factor in the progenitors at late stages of cerebral cortex development.

They discover that Foxg1 is a repressor of gliogenesis and has a dual function, first as a repressor of Fgfr3 receptor in progenitors, and second as a suppressor of the Fgf ligands in young neurons.

They found that the inactivation of Foxg1 in cortical progenitors causes premature astrogliogenesis at the expense of neurogenesis. They identify Fgfr3 as a novel FOXG1 target. They show that suppression of Fgfr3 by FOXG1 in progenitors is required to maintain neurogenesis. On the other hand, they also show that FOXG1 negatively regulates the expression of Fgf gliogenic secreted factors in young neurons suppressing gliogenesis cells extrinsically.

Strengths:

The authors used time-consuming in vivo experiments utilizing several mouse strains including Foxg1-MADM in combination with RNA-Seq and ChIP to convincingly show that Foxg1 acts upstream of FGF signalling in the control of gliogenesis onset. The conclusions of this paper are mostly well supported by data.

---

## [Referee Report · Reviewer #2 (Public review)]

Summary:

We have known for some time that neural progenitors in the cerebral cortex switch their output from cortical neurons to glia at late embryonic stages, however little is known about how this switch is regulated at the molecular level. Bose et al present a convincing set of findings, demonstrating that the transcription factor Foxg1 plays a key role in this process, mediated through FGF signalling. Foxg1 cell-autonomously inhibits gliogenesis in progenitor cells (thereby promoting neuronal identity), and lower Foxg1 expression in postnatal neurons leads to increased expression of FGF ligand, promoting glial development from nearby progenitors.

Strengths:

The study is very well designed, having a systematic, thorough, and logical approach. The data is convincing. The authors make full use of a range of existing transgenic strains, published 'omics data, and elegant genetic approaches such as MADM. This combination of approaches is particularly rigorous, lending significant weight to the study. The manuscript is well-written, clear, and easy to follow.

Impact

This manuscript identifies a previously unknown role for Foxg1 in forebrain development and a mechanism underlying the neurogenic-to-gliogenic switch that occurs at late embryonic stages of cortex development. These findings will stimulate further research to uncover more details of how this important switch is controlled and may provide useful insight into some of the symptoms experienced by children with FOXG1 Syndrome.

---

## [Author Response]

The following is the authors’ response to the original reviews.

**Weaknesses (Reviewer 1):**
The role of Fgf signaling in gliogenesis and Foxg1 in neurogenesis is well known. It is not clear if Fgf18 is a direct target of Foxg1.

We agree with the reviewer- Fgf signaling is an established pro-gliogenic pathway (Duong et al 2019) and *Foxg1* overexpression is known to promote neurogenesis in cultured neural stem cells (Branacaccio et al 2019). Our study links these two mechanisms, as the Reviewer has summarized: (a) we demonstrate that FOXG1 works via modulating Fgf signaling cell-autonomously within progenitors by regulating the levels of Fgfr3. (b) Loss of *Foxg1* in postmitotic neurons results in the upregulation of Fgf ligand expression (possibly via indirect mechanisms) and this non-cell autonomously increases Fgf signaling in progenitors_._ Our study is entirely performed in vivo*.*

Revision: We have revised the manuscript to reflect that *Fgf18* may be an *indirect* target of FOXG1 in postmitotic neurons.

**Weaknesses (Reviewer 2):**
It wasn't clear to me why the authors chose postnatal day 14 to examine the effects of Foxg1 deletion at E15 - this is a long time window, giving time for indirect consequences of Foxg1 deletion to influence development and thereby potentially complicating the interpretation of findings. For example, the authors show that there is no increased proliferation of astrocytes or death of neurons lacking Foxg1 shortly after cre-mediated deletion, but it remains formally possible (if perhaps unlikely) that these processes could be affected later during the time window. The rationale underlying the choice of this time point should be explained.I don't agree with the statement in the very last sentence of the results section that "neurogenesis is not possible in the absence of [Foxg1]" as there are multiple reports in the literature demonstrating the presence of neurons in Foxg1-/- mice (eg: Xuan et al., 1995; Hanashima et al., 2002, Martynoga et al., 2005, Muzio and Mallamaci 2005). Perhaps the statement refers specifically to late-born cortical neurons. This point also arises in the discussion section.

Revisions:

(a) We have revised the manuscript to explain why we chose postnatal day 14 to examine the effects of Foxg1 deletion at E15.

We have examined the transcriptomic dysregulation after *Foxg1* deletion at E17.5, which is a reasonable period to identify potential direct targets. Furthermore, FOXG1 occupies the *Fgfr3* locus in ChIP-seq performed at E15.5. Together, these support the interpretation that *Fgfr3* is a direct target of Foxg1.As the Reviewer notes, we have investigated the possibility of increased proliferation of astrocytes and death of neurons and found no evidence suggesting these phenomena occur in the 3 days after loss of Foxg1. Cortical neurons are postmitotic and differentiated by E18.5, the stage at which we examined CC3 staining and found no difference in cell death in control and mutants (Supplementary Figure S2C, C’). The majority of progenitors (PAX6+ve cells) that lose Foxg1 at E15.5 express the gliogenic transcription factor NFIA by E18.5 (Figure 2C, C’), but hardly any express intermediate (neurogenic) progenitor marker TBR2 (Supplementary Figure S2B, B’). It is therefore unlikely that neurons are born from Foxg1 mutant progenitors and then die at a later stage.The cellular consequences of loss of Foxg1 require additional time to detect e.g. it takes ~ 5 days for GFAP to be detected in astrocytes once they are born. The P14 timepoint permits the assessment of oligogenesis which begins after astrogliogenesis and therefore permits a comprehensive assessment of the lineage of E15.5 Foxg1 null progenitors.

(b) Thank you for pointing out that the last sentence of the results section implied (incorrectly) that ALL neurogenesis is not possible in the absence of Foxg1 We have modified this (and the discussion) to reflect that this applies to E14/15 progenitors and late-born cortical neurons.

**Recommendations for the authors (Reviewer 2):**

(c) We thank the reviewer for this suggestion. We will modify the schematic (Figure 7) to remove any ambiguity regarding Foxg1 expression.